# EarlySciRev: Un jeu de données sur les révisions scientifiques à un stade précoce, extraites des traces d'écriture LaTeX

Léane Jourdan[1]    Julien Aubert-Béduchaud[1]    Yannis Chupin[1]    Marah Baccari[1]
Florian Boudin[2]

(1) Nantes Université, École Centrale Nantes, CNRS, LS2N, UMR 6004, F-44000 Nantes, France
(2) Inria, LS2N, Nantes Université, France
`prenom.nom@univ-nantes.fr`

## RÉSUMÉ

La rédaction scientifique est un processus itératif qui génère de nombreuses traces de révision, mais les ressources publiquement accessibles ne présentent généralement que les versions finales ou quasi finales des articles. Cela limite l'étude empirique des comportements de révision et l'évaluation des grands modèles de langue (LLM) pour la rédaction scientifique. Nous présentons EarlySciRev, un jeu de données de révisions de textes scientifiques à un stade précoce, extraites automatiquement des fichiers source LaTeX d'arXiv. Notre observation clé est que le texte commenté en LaTeX conserve souvent des formulations rejetées ou alternatives rédigées par les auteurs eux-mêmes. En alignant les segments commentés avec le texte final adjacent, nous extrayons des paires de révisions candidates au niveau du paragraphe et appliquons un filtrage basé sur les LLM pour conserver les révisions réelles. À partir de 1,28 million de paires candidates, notre pipeline extrait 578 000 paires de révisions validées, fondées sur des traces authentiques des premières ébauches. Nous fournissons en outre un benchmark annoté par des humains pour la détection des révisions.

## ABSTRACT

**EarlySciRev : A Dataset of Early-Stage Scientific Revisions Extracted from LaTeX Writing Traces**

Scientific writing is an iterative process that generates rich revision traces, yet publicly available resources typically expose only final or near-final versions of papers. This limits empirical study of revision behaviour and evaluation of large language models (LLMs) for scientific writing. We introduce EarlySciRev, a dataset of early-stage scientific text revisions automatically extracted from arXiv LaTeX source files. Our key observation is that commented-out text in LaTeX often preserves discarded or alternative formulations written by the authors themselves. By aligning commented segments with nearby final text, we extract paragraph-level candidate revision pairs and apply LLM-based filtering to retain genuine revisions. Starting from 1.28M candidate pairs, our pipeline yields 578k validated revision pairs, grounded in authentic early drafting traces. We additionally provide a human-annotated benchmark for revision detection. EarlySciRev complements existing resources focused on late-stage revisions or synthetic rewrites and supports research on scientific writing dynamics, revision modelling, and LLM-assisted editing.

MOTS-CLÉS : jeu de données, révision de texte, articles scientifiques, filtrage par LLM.

KEYWORDS: dataset, text revision, scientific articles, LLM filtering.

ARTICLE ACCEPTÉ À : 3rd International Workshop on Natural Scientific Language Processing.

# 1 Introduction

La rédaction scientifique est une tâche intrinsèquement exigeante, qui requiert précision, clarté et concision, souvent dans l'urgence et fréquemment dans une langue étrangère. Pour faciliter ce processus, les chercheurs s'appuient de plus en plus sur les grands modèles de langue (LLM), capables de réécrire des passages entiers en réponse à des instructions de haut niveau. Ces modèles offrent une aide sans précédent pour réviser les brouillons, améliorer la lisibilité et exprimer des idées complexes avec clarté et aisance. Cependant, évaluer l'impact des LLMs sur la rédaction scientifique reste un défi. Les données clés nécessaires à une telle analyse, à savoir les *brouillons, les révisions et les traces de rédaction*, sont en grande partie inaccessibles. Si la plupart des articles scientifiques sont accessibles au public sous leur forme finale ou quasi finale, le processus de rédaction itératif qui les a produits reste généralement caché.

La rédaction scientifique étant par nature un processus itératif, les auteurs affinent progressivement leurs arguments, restructurent leurs paragraphes, clarifient leurs explications et ajustent leurs affirmations. Ce processus génère naturellement divers artefacts intermédiaires, notamment des phrases écartées, des paragraphes réécrits et des alternatives commentées. Pourtant, ces traces sont rarement conservées dans des ressources accessibles au public. En conséquence, les recherches existantes sur la modélisation de la révision s'appuient principalement sur des révisions de dernière minute (par exemple, entre les versions soumises) ou sur des réécritures synthétiques. Les premières étapes de rédaction, au cours desquelles des changements conceptuels et structurels importants se produisent, restent sous-explorées.

L'impossibilité d'accéder aux révisions effectuées aux premières étapes d'écriture limite notre capacité à étudier les dynamiques réelles de la rédaction, à mesurer les améliorations en termes de qualité, à entraîner des modèles permettant une révision approfondie et à évaluer le rôle des LLMs dans la rédaction scientifique. Cela entrave également l'étude systématique de problèmes tels que l'homogénéisation stylistique, la propagation des biais ou la déformation des faits introduits lors de la réécriture automatisée.

Cette inaccessibilité tient en grande partie à des considérations d'ordre éthique, juridique et de propriété : les brouillons peuvent contenir des informations personnelles, des idées non publiées ou des commentaires confidentiels, et elles se rapportent souvent à des articles qui relèvent par la suite des droits d'auteur de l'éditeur.

Dans cet article, nous présentons EarlySciRev, un jeu de données à grande échelle contenant des révisions scientifiques à un stade précoce, extraites automatiquement des fichiers source LaTeX d'arXiv. Notre observation clé est que les commentaires LaTeX conservent souvent des versions rejetées ou intermédiaires de phrases et de paragraphes. En exploitant ces segments commentés et en les alignant avec le texte final adjacent, nous récupérons des paires de révisions détaillées qui reflètent les modifications authentiques apportées par l'auteur. Nous présentons un pipeline complet [1] pour (i) collecter des articles en informatique sur arXiv, (ii) nettoyer et traiter les sources LaTeX, (iii) extraire des paires de révisions candidates à partir du texte commenté, et (iv) filtrer les révisions réelles à l'aide d'une classification basée sur un LLM. Enfin, nous présentons une étude d'annotation

---

1. https://github.com/JourdanL/EarlySciRev

qui compare plusieurs LLMs sur la tâche de détection des révisions, et utilisons le meilleur modèle pour filtrer le jeu de données final.

Notre contribution sont :

— Nous proposons une nouvelle méthode permettant de récupérer les traces des premières étapes de la rédaction scientifique en exploitant les commentaires présents dans les fichiers source LaTeX.

— Nous présentons EarlySciRev, un jeu de données à grande échelle contenant des révisions scientifiques au niveau des paragraphes, extraites automatiquement d'articles publiés sur arXiv, qui reflètent fidèlement les premières ébauches de révision. [2]

— Nous publions un jeu de données de référence annoté par des humains pour la détection des révisions dans les textes scientifiques, permettant une évaluation systématique à la fois des LLMs et des futurs modèles de révision.[2]

## 2   État de l'art

Au fil des années, divers jeux de données consacrés à la révision de textes ont été publiés, reflétant l'intérêt croissant pour la modélisation des processus d'écriture et de réécriture. Certains de ces jeux de données s'appuient sur des révisions synthétiques, générées soit automatiquement (Ito *et al.*, 2019), soit manuellement par des annotateurs qui ne sont pas les auteurs originaux (Mita *et al.*, 2024). Si ces ressources sont utiles pour des expérimentations contrôlées, elles ne reflètent pas le comportement réel des auteurs en matière de révision.

Parmi les jeux de données contenant de véritables révisions d'auteurs, deux sources principales se sont imposées : arXiv (Tan & Lee, 2014; Du *et al.*, 2022; Jiang *et al.*, 2022) et OpenReview (D'Arcy *et al.*, 2023; Jourdan *et al.*, 2024, 2025). Ces ressources permettent d'étudier les comportements de révision dans des contextes réels de rédaction scientifique.

Malgré leur utilité, les jeux de données existants présentent des limites claires. Tout d'abord, beaucoup ont une portée limitée. Certains se concentrent exclusivement sur les résumés (Tan & Lee, 2014; Du *et al.*, 2022), tandis que d'autres restent d'une taille relativement modeste, allant de quelques centaines (Jiang *et al.*, 2022; Mita *et al.*, 2024; Ruan *et al.*, 2024) à quelques milliers d'articles (Kuznetsov *et al.*, 2022; D'Arcy *et al.*, 2023; Dycke *et al.*, 2023; Lin *et al.*, 2023; Jourdan *et al.*, 2025).

Deuxièmement, la plupart des jeux de données existants capturent principalement les révisions de dernière minute, généralement celles effectuées entre les versions quasi-finales publiées sur arXiv et les plateformes de soumission. Ces révisions reflètent souvent des manuscrits déjà peaufinés, prêts à être diffusés au public, plutôt que les étapes exploratoires et formatives de la rédaction. Les premières phases de rédaction, au cours desquelles des modifications conceptuelles, structurelles et stylistiques substantielles sont apportées, sont donc largement absentes des ressources actuelles. À notre connaissance, le seul jeu de données qui cible explicitement les traces d'écriture dès les premières étapes du processus de rédaction est ScholaWrite (Wang *et al.*, 2025). Cependant, il se limite à seulement cinq articles, ce qui restreint son applicabilité au-delà de l'analyse exploratoire.

En conséquence, l'accès aux traces d'écriture des premières étapes reste un obstacle majeur à l'étude des comportements réels de révision scientifique et à l'élaboration de modèles permettant

---

une révision approfondie. Dans ce travail, nous émettons l'hypothèse que ces traces peuvent être récupérées directement à partir des fichiers source LaTeX téléchargés sur arXiv. Dans la pratique, les auteurs laissent fréquemment dans le code source des phrases, des paragraphes ou des formulations alternatives commentées (i.e. des lignes commençant par le caractère « % »). L'exploration de ces segments commentés offre une voie prometteuse et peu explorée pour reconstruire les révisions scientifiques précoces.

# 3 Création des données

Cette section décrit le processus utilisé pour construire le jeu de données, depuis la collecte des données brutes sur arXiv jusqu'au nettoyage des sources LaTeX et à l'extraction des paires de révisions candidates.

## 3.1 Collecte des données

Nous nous appuyons sur le fichier `arxiv-metadata-oai-snapshot.json`[3] téléchargé le 21 janvier 2026. Ce fichier de métadonnées contient 2 932 928 dépôts arXiv, dont 596 118 sont distribués sous une licence permissive.[4]

Dans le cadre de ce travail, nous nous concentrons sur les articles en informatique (CS), dont 286 747 sont assortis d'une licence valide. Pour chacun de ces articles, nous avons téléchargé toutes les archives sources disponibles (fichiers zip), ce qui a généré environ 1,2 To de données sources.

## 3.2 Traitement des sources LaTeX

Nous filtrons d'abord les fichiers sources afin de ne conserver que les documents LaTeX contenant des commentaires potentiellement exploitables. Plus précisément, nous sélectionnons les fichiers qui contiennent des lignes commentées ne commençant *pas* par une barre oblique inversée. Ce critère exclut les commandes LaTeX mises en commentaire, tout en conservant les commentaires susceptibles de contenir des versions antérieures de phrases ou de paragraphes.

Nous appliquons ensuite les étapes de nettoyage suivantes :

1. Nous ne conservons que le contenu situé entre `\begin{document}` et `\end{document}`.
2. Nous supprimons les environnements non textuels, notamment `table`, `figure`, `align`, `tikz` et `algorithm`.
3. Nous remplaçons chaque environnement `equation` par un token spécial `[EQUATION]`, car des équations peuvent apparaître dans le corps du texte.
4. Nous supprimons les commandes LaTeX qui ne contiennent pas de contenu textuel (par exemple, `\appendix`, `\vspace`), tout en conservant les commandes structurelles telles que `\section`.

Ces étapes garantissent que le texte obtenu se compose principalement d'un contenu en langage naturel adapté à l'analyse de révision.

---

3. https://www.kaggle.com/datasets/Cornell-University/arxiv
4. CC BY-NC-SA 4.0, CC BY-SA 4.0, CC BY 4.0, Public Domain List, CC BY-NC-SA 3.0, CC BY 3.0, CC0 1.0

## 3.3 Extraction de paires de révisions de candidats

Notre objectif est d'extraire des paires de révisions candidates composées : i) d'un bloc de texte non commenté figurant dans le document compilé (le *paragraphe final*), et (ii) d'un bloc de texte commenté pouvant correspondre à une version antérieure (le *paragraphe commenté*). Nous définissons un *bloc* comme une séquence de lignes consécutives du même type (commentées ou non commentées), non interrompue par une ligne vide ou par une ligne de l'autre type.

Pour chaque bloc commenté $c$, nous le comparons aux cinq blocs précédents et aux cinq blocs suivants. Pour chaque bloc voisin correspondant à un paragraphe final $f$, nous calculons un ratio de différence normalisé basé sur la distance de Levenshtein entre les deux textes. Afin de prendre en compte les cas où une révision ne concerne qu'une partie d'un paragraphe, nous calculons cette distance à l'aide d'une fenêtre glissante sur le paragraphe final. Nous définissons ce ratio de différence normalisé comme suit :

$$d_{norm}(f, c) = lev(f, c) / \max(|f|, |c|) \tag{1}$$

où $lev(\cdot, \cdot)$ désigne la distance de Levenshtein et $|\cdot|$ la longueur en caractères. Nous considérons une paire comme une révision candidate lorsque $d_{norm} < 0,7$. Ce seuil a été fixé de manière empirique sur un sous-ensemble du jeu de données.

Pour chaque paragraphe final, nous conservons toutes les révisions commentées candidates associées. Avant de concaténer les blocs non commentés, nous supprimons les commentaires en fin de ligne afin de garantir que le texte obtenu corresponde au contenu du document compilé.

# 4 Annotation des données

La procédure d'extraction automatique décrite ci-dessus a permis d'identifier 1 269 976 paires de révisions candidates. Cependant, toutes ces paires candidates ne correspondent pas à de véritables cas de réécriture. Nous introduisons donc une étape de filtrage supplémentaire basée sur les LLMs. Afin de choisir une stratégie de prompt et un modèle appropriés, nous construisons d'abord un sous-ensemble de référence annoté par des humains.

## 4.1 Annotation humaine

L'objectif de cette campagne d'annotation est de déterminer si une paire de paragraphes donnée constitue un véritable cas de révision. Nous avons sélectionné au hasard 500 paires candidates, chacune composée d'un *paragraphe commenté* (représentant une version originale potentielle) et d'un *paragraphe final*. L'annotation a été réalisée par cinq annotateurs : deux étudiants en master, deux chercheurs juniors et un chercheur senior. Tous les annotateurs étaient non-natifs de l'anglais et avaient une expérience préalable de la rédaction scientifique et de la recherche en TAL.

Pour chaque paire de paragraphes, les annotateurs ont répondu à la question binaire suivante : *« Le paragraphe final peut-il être qualifié de révision du ou des paragraphes originaux ? »* Les annotateurs ont reçu un guide détaillé précisant les critères de décision (voir l'annexe A). Ce guide est dérivé de la taxonomie de révision proposée par Jourdan *et al.* (2025) et adaptées à la tâche.

L'annotation a été réalisée à l'aide de *Label Studio*. Les paires de paragraphes ont été affichées côte à

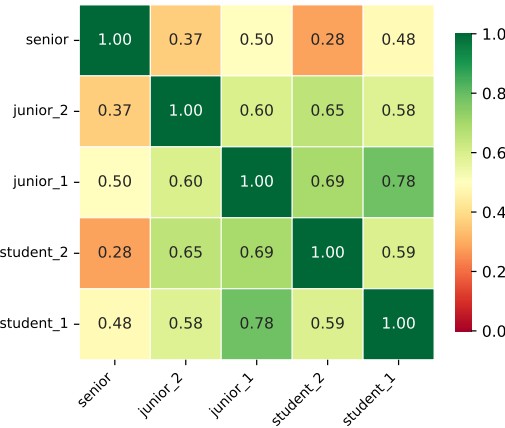

FIGURE 1 – Pairwise Cohen's Kappa ($\kappa_{\text{Cohen}}$) scores between annotators.

côte. Pour faciliter la comparaison, les segments de texte communs aux deux paragraphes ont été mis en évidence (annexe B).

**Accord inter-annotateurs.**

La figure 1 présente les Kappa de Cohen $\kappa_{\text{Cohen}}$ par paires entre les annotateurs. Le degré d'accord varie selon les paires d'annotateurs, un accord moindre étant observé pour les paires incluant l'annotateur senior. L'annotateur senior s'est montré plus sélectif que les autres annotateurs sur les paires considérées comme nécessitant une révision.

Pour évaluer l'accord global entre les annotateurs, nous utilisons le Kappa de Fleiss $\kappa_{\text{Fleiss}}$ dans le cadre d'un protocole à chevauchement partiel, dans lequel chaque élément est annoté par trois annotateurs choisis parmi le groupe de cinq. Le $\kappa_{\text{Fleiss}}$ obtenu, soit 0,54, indique un accord modéré, selon l'échelle proposée par Landis & Koch (1977).

## 4.2 Filtrage basé sur les LLM

Compte tenu du nombre de paires de candidats extraites, l'annotation manuelle du jeu de données complet est irréalisable. Nous nous appuyons donc sur la classification automatique pour identifier les paires de révisions réelles. Afin de sélectionner l'approche la plus fiable, nous évaluons plusieurs modèles et stratégies d'amorce sur le sous-ensemble annoté par des humains. Étant donné que chaque élément de ce sous-ensemble a été étiqueté par trois annotateurs, nous utilisons le vote majoritaire pour déterminer l'étiquette de référence pour l'évaluation.

**Approches.** Nous comparons deux approches basées sur les LLM : i) **Amorçage standard (LLM-choix)**, où le modèle répond directement à la question binaire posée dans la requête, et la décision est extraite de la sortie générée. ii) **\*-PLUIE** (Lemesle *et al.*, 2026), une méthode \*LLM-as-a-judge\* basée sur la perplexité qui estime la confiance du modèle sans générer de texte libre. Dans le cadre de \*-PLUIE, le modèle estime sa confiance en calculant la perplexité des réponses candidates (« Yes » vs « No ») compte tenu de l'amorce. Les valeurs positives favorisent l'hypothèse de révision, les valeurs négatives l'inverse, et une décision binaire est obtenue en appliquant un seuil au score (le seuil par

FIGURE 2 – Amorce utilisée à la fois dans les configurations *-PLUIE et LLM-choix.

défaut est 0 mais il peut être optimisé a posteriori lorsque des annotations sont disponibles).

**Modèles et configuration.** Nous évaluons plusieurs LLMs de tailles variées, notamment Qwen3 (4B,14B), phi-4 (14B), Olmo-3-7B-Instructet Llama-3.1-8B-Instruct. Tous les modèles sont exécutés dans un contexte de complétion de conversation avec une précision `bfloat16`. L'échantillonnage est désactivé pendant l'inférence (`do_sample=False`) afin de garantir des résultats déterministes. La même amorce (Figure 2) est utilisée à la fois pour les configurations standard et *-PLUIE, ce qui permet une comparaison directe entre la classification basée sur la génération et celle basée sur la perplexité pour chaque modèle.

**Résultats et sélection du modèle.**

Les résultats sont présentés dans la Table 1. Dans les deux approches, Qwen3 (14B) obtient la meilleure performance globale, avec une précision supérieure à 80%. Afin de trouver un équilibre entre la performance de classification et l'efficacité computationnelle, nous sélectionnons la configuration *-PLUIE avec Qwen3 (14B) pour filtrer le jeu de données complet, en utilisant le seuil optimal déterminé sur le sous-ensemble annoté.

# 5  Statistiques du jeu de données

L'application de la stratégie de filtrage sélectionnée a permis de retenir 578 440 paires de révisions sur les 1,2 million de candidates initiales (environ 45,55%). Ces révisions validées correspondent à 523 932 paragraphes finaux distincts. Parmi ceux-ci, 46 192 paragraphes sont associés à plus d'une révision candidate, ce qui reflète les cas où les auteurs ont testé plusieurs formulations alternatives avant de se décider pour une version finale, ou les cas où plusieurs paragraphes précédents ont été fusionnés en un seul. Au niveau des documents, les révisions sont réparties sur 104 023 articles. Ces caractéristiques, ainsi que d'autres, sont résumées dans la Table 2.

L'examen qualitatif montre que les modifications enregistrées dans EarlySciRev vont de simples corrections de style à des restructurations et clarifications plus importantes des arguments scientifiques, en passant par des brouillons d'idées peu rédigées et leur version définitive.

Les 500 paragraphes utilisés pour l'annotation humaine sont également inclus et filtrés à cette étape. Le sous-ensemble de données annoté par des humains et celui filtré par un LLM sont librement accessibles.

| | Model | Acc. | P. | R. | time |
|---|---|---|---|---|---|
| **\*-PLUIE** | Qwen3 (4B) (thr=0) | 0.77 | 0.77 | 0.76 | 22h |
| | Qwen3 (4B) (thr=-4.75) | 0.78 | 0.75 | 0.81 | |
| | Olmo-3 (7B) (thr=0) | 0.74 | 0.73 | 0.76 | 35h |
| | Olmo-3 (7B) (thr=0.60) | 0.74 | 0.78 | 0.67 | |
| | Llama-3.1 (8B) (thr=0) | 0.70 | 0.65 | 0.85 | 35h |
| | Llama-3.1 (8B) (thr=0.85) | 0.73 | 0.75 | 0.67 | |
| | phi-4 (14B) (thr=0) | 0.77 | 0.71 | 0.92 | 55h |
| | phi-4 (14B) (thr=2.15) | 0.79 | 0.75 | 0.85 | |
| | Qwen3 (14B) (thr=0) | 0.80 | 0.75 | 0.90 | 62h |
| | Qwen3 (14B) (thr=5.55) | **0.82** | 0.80 | 0.86 | |
| **LLM-choice** | Qwen3 (4B) | 0.59 | 0.55 | 0.95 | 29h |
| | Llama-3.1 (8B) | 0.59 | 0.55 | **0.97** | 46h |
| | Olmo-3 (7B) | 0.68 | **0.81** | 0.45 | 45h |
| | phi-4 (14B) | 0.78 | 0.80 | 0.75 | 81h |
| | Qwen3 (14B) | 0.80 | 0.78 | 0.83 | 82h |

TABLE 1 – Alignement d'un classificateur basé sur un LLM avec le vote majoritaire humain. *Seuil* est le seuil utilisé pour binariser les valeurs \*-PLUIE, *Acc.* l'exactitude, *P.* la précision, *R.* le rappel et *temps* le temps estimé pour classer toutes les données. Les valeurs en **gras** indiquent les meilleurs résultats, et les valeurs soulignées indiquent les deuxièmes meilleurs résultats.

| #rev | #paper | #rev/§ | #words/§ | % words diff |
|---|---|---|---|---|
| 578,440 | 104,023 | 1.10 | 82.42 | 56.85 |

TABLE 2 – Caractéristiques de EarlySciRev. Dans cet ordre : nombre de paires de révisions, nombre d'articles, nombre moyen de paragraphes commentés par paragraphe final, nombre moyen de mots par paragraphe (version finale), pourcentage moyen de différence en nombre de mots par paire de révisions

# 6 Conclusion

Nous avons présenté EarlySciRev, un jeu de données regroupant les révisions scientifiques au niveau des paragraphes extraites des traces d'écriture LaTeX contenues dans les fichiers sources d'arXiv, ainsi qu'un jeu de référence annoté par des humains pour la détection des révisions. En se concentrant sur les premières étapes de la rédaction, cette ressource met en évidence des phénomènes de révision qui sont généralement invisibles dans les jeux de données existants.

EarlySciRev permet l'étude empirique des dynamiques de la rédaction scientifique et sert de base à l'élaboration et à l'évaluation de modèles de révision, dont les systèmes basés sur des LLMs. Afin de faciliter la reproductibilité et de futurs développements, nous mettons à disposition l'intégralité du cadre d'extraction et de filtrage, ce qui permet d'effectuer des mises à jour à mesure que de nouveaux articles sont publiés.

Une prochaine étape pourrait consister à annoter toutes les données en indiquant l'intention de révision, puis à comparer la répartition obtenue avec celle d'un jeu de données axé sur les révisions en phase finale.

# 7    Limites

Le pipeline actuel se limite aux articles en informatique, car nous ne traitons que les articles CS sous licence provenant d'arXiv. Les pratiques rédactionnelles et les comportements en matière de révision peuvent varier d'une discipline à l'autre, ce qui limite la généralisation de nos résultats. L'extension de cette approche à d'autres domaines constitue une orientation naturelle pour nos travaux futurs.

De plus, nous ne contrôlons pas explicitement la langue de l'article. Une partie des articles soumis sur arXiv sont rédigés dans des langues autres que l'anglais. Notre filtrage basé sur un LLM repose sur une amorce rédigée en anglais, ce qui peut affecter la fiabilité de la classification pour les textes non anglophones. Adapter l'amorce à la langue de chaque document ou intégrer l'identification de la langue dans le pipeline pourrait améliorer la robustesse du système.

Enfin, bien que des révisions itératives soient présentes dans le jeu de données, sans indications temporelles liées aux modifications, il n'est pas possible de les ordonner et d'exploiter cette information.

# Remerciements

This work was partly supported the AID-CNRS NaviTerm project (convention 2022 65 0079 CNRS Occitanie Ouest).

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

# A   Guide d'annotation

**Guide d'annotation pour la détection de révision**

**1. Introduction**

The goal of this annotation campaign is to detect text revisions amid paragraphs originating from computer science scientific papers. A paragraph level revision is defined as a paragraph that is substantially modified for clarity, simplicity, style and other aspects. To that end, some final paragraphs have been selected and each one of them was provided with one or more original paragraphs that were under comment in the latex file. A final paragraph is a paragraph that is not commented and is suspected to be a revision of an original paragraph(s). In this task, we aim to characterize the final paragraphs' relationship with the suspected original paragraph(s), so that they can be classified as revisions or not down the line.

**2. Annotation Task**

Annotators are presented with a pair of paragraphs : an original version composed of one or several paragraphs and a final version. Their task is to answer the following question : *Can the final paragraph be qualified as a revision of the original one(s) ?*

Annotators must select one of the following labels :

— **YES** : The final paragraph constitutes a revision of the original paragraph.

— **NO** : The final paragraph does not constitute a revision (e.g., different scientific content, the idea developed is not the same, introduces too much new information, or does not change the text).

As several original candidates are proposed, the annotator can answer Yes for multiple paragraphs (e.g. in cases of paragraph merging or iterative revision).

**2.1 Positive example**

| Original Paragraph | Final paragraph |
|---|---|
| Therefore, the generalization rapidly decreases after augmentationinterrupted when training with a single background because the learning direction toward generalization about various backgrounds is not helpful to train. On the other hand, the training can have helpwhen their difculty is solved by augmentation, such as Figure 2(b) and Figure 2(c). | Therefore, the generalization rapidly decreases after augmentation is interrupted during training with a single background because the learning direction toward generalization about various backgrounds is not helpful to train. In contrast, the training can help when their difficulty is solved by augmentation (Figure 2(b), 2(c)). |

**2.2 Negative example**

| Original Paragraph | Final paragraph |
|---|---|
| In future research, the multi-mode characteristics will be studied to improve the representativeness of degradation features and the trendability of HI, and transfer learning approaches will be investigated to improve the generalization ability of the proposed framework and extend it to different systems. | Based on the ablation study, it can be concluded that the proposed SkipAE, inner HI-prediction block, and the HI-generating module jointly improve the ability of HI for reliable and accurate prognostics. |

**2.3 Annotation Procedure**

For each pair of paragraphs (original and final), annotators must proceed as follows :

1. Read the final paragraph carefully to understand its scientific content and intent.

2. Read the original paragraph to identify any differences with respect to the final version.

3. Assess whether each original is rephrased in the final paragraph considering aspects such as : grammatical correctness, clarity and readability, fluency and coherence, appropriateness of scientific style.

4. Determine whether the scientific meaning of the paragraph is preserved in the final version.

5. Assign a label (YES or NO) according to the decision rules defined below.

Annotators should base their decision solely on the information contained in the paragraph pair and should not rely on external context. Also, annotators are prohibited to invent things.

**2.4 Decision Rules**

Annotators must apply the following rules when assigning labels :

**Assign YES** if at least one of the following conditions are met for a part or the whole paragraph :

1. The final paragraph is a revised version of the original paragraph, incorporating changes ranging from minor edits to substantial rephrasing.

2. The final version has been modified through the addition, the substitution or the deletion of ideas or facts.

3. The revision expands on the same idea with additional or withdrawn details.

4. The differences between the original and final paragraphs indicate the correction of document processing errors (e.g., parsing issues, segmentation errors, or misaligned paragraphs).

**Assign NO** if :

1. None of the above conditions are met.

2. If the annotator is unsure whether the revised paragraph constitutes a valid paragraph-level revision.

3. If there are only equations or code.

4. If the two paragraphs are the exact same.

If presented with multiple commented paragraphs for the same final paragraph, one or more commented paragraph can independently be considered as a revision. Classifying a commented paragraph as a revision does not disqualify the other proposed candidates. The same goes with the negative label : all the commented paragraphs may not qualify as a revision.

**Table 3 :** Guide détaillé fourni aux annotateurs pour la tâche de détection des révisions.

# B Exemples d'annotations

| Référence | Candidat | Révisé |
|---|---|---|
| We observed that AR2VP demonstrates superior entity perception outcomes, achieving the highest overall perception performance. This analysis underscores that current V2X technologies rarely rely on RSUs to expand perception horizons. In contrast, AR2VP harnesses the latent strengths of RSUs to address intra-scene changes, which enhances the vehicle's ability to adapt to dynamic scenes, consequently elevating the overall perception capabilities. However, AR2VP does exhibit a performance drawback in pedestrian detection, implying a particular challenge in detecting small targets. | We find that AR2VP shows superior entity perception results, with overall perception performance being the best. This analysis suggests that existing V2X technologies merely utilize RSU to extend perception horizons. In contrast, AR2VP leverages the latent advantages of RSU to model intra-scene changes, further enhancing vehicle adaptability to scene dynamics, thereby augmenting overall perception capabilities. However, in comparison to V2V, AR2VP exhibits a performance disadvantage in pedestrian detection, indicating a certain discrepancy in detecting small targets. | Oui |
| Under \cref{ass :runtimecomplexity} and following \cref{tab :setops}, the outer approximative Minkowski sum from \cref{prop :minkSum_polyzono}, the Minkowski difference, and the linear map in the computation of the outer approximation $\outerBRSAE{-t}$ are all $\bigO{n^3}$, while the computation of the inner approximation $\innerBRSAE{-t}$ is dominated by the conversion to a constrained zonotope, which is $\bigO{n^4}$. | Under \cref{ass :runtimecomplexity}, the computation of the outer approximation $\outerBRSAE{-t}$ is marginally dominated by the over-approximative Minkowski sum, which is $\bigO{(\cons{}+2n)n$, since the Minkowski difference and linear map are at most $\bigO{(\cons{}+2n)n \steps{}n}$ and $\bigO{(\cons{}+2n)n^2}$, respectively ; all these operations are essentially $\bigO{n^3}$ for $n \gg \steps{}$ and under \cref{ass :runtimecomplexity}. | Oui |
| $$\sum_{t=1}^T \sum_{y \in \{0, 1\}} p_t^y \hat{\ell}_t(y) - \inf_{j \in [N]}\sum_{t=1}^T \hat{\ell}_t(\mathcal{E}_t^{j} ) \leq \frac{\ln N}{\eta} + \eta \sum_{t=1}^T \hat{\ell}_t(1) + \eta\sum_{t=1}^T p_t^1(1 - p_t^1) \hat{\ell}_t(0)^2 + \eta\sum_{t=1}^T p_t^1 \hat{\ell}_t(1)^2.$$ \end{lemma} | $$\sum_{t=1}^T \sum_{y \in \{0, 1\}} p_t^y \hat{\ell}_t(y) - \sum_{t=1}^T \sum_{y \in \{0, 1\}} \hat{\ell}_t(\mathcal{E}_t^{j}) ) \leq \frac{\ln N}{\eta} + \eta \sum_{t=1}^T \hat{\ell}_t(1) + \eta\sum_{t=1}^T \sum_{y \in \{0, 1\}} p_t^y \hat{\ell}_t(y)^2.$$ | Non |
| \iuhead{Date of fault-triggering test creation and modification :} We identified all the commits that are associated with the fault-triggering tests and analyzed when the commits happened (e.g., before or after the bug was reported/fixed). We used the git command "\inlineCode{git log -L :[funcname] :[file]}" to identify the list of commits that modified the fault-triggering test and the modification date. | We collected the date and time information provided in the output of this command to track the development activities associated with each fault-triggering test. This allowed us to better understand how the fault was identified and resolved over time with respect to the changes in fault-triggering tests. \peter{maybe remove this, I don't quite get this detail}Note that if a test is inherited from a parent class, we perform our analysis directly on the parent class since any changes would only occur in that class. | Non |

**Table 4 :** Exemple de paires de paragraphes annotés ; les segments de texte identiques sont indiquées en vert.