# OpenReview forum: "EarlySciRev : A Dataset of Early-Stage Scientific Revisions Extracted from LaTeX Writing Traces"
_ls2n.fr/CORIA-TALN/2026/Workshop/ARTS — ls2n CORIATALN 2026 Workshop ARTS Submission_

### Official Review · Reviewer_TxMG · 2026-05-05

**Mode De Presentation:** Oral

**Confience:**

Oui

**Decision:**

Accepté

**Relecture:**

L'article décrit un jeu de données intéressant puisqu'assez original
et ouvrant des perspectves de recherche intéressantes.
La démarche est décrite en détail.

Il serait néanmoins intéressant d'avoir des détails sur le processus
d'annotation. Ainsi, en supposant que les textes soient en anglais,
quel est leur niveau d'anglais. Quelle était la décision à prendre
lorsque le texte n'était pas en anglais (ou en français) ?

Il n'est pas clair si la séquentialité des révisions (R1->R2->R3) est
identifiée et pris en compte dans la classification par les modèles.

Les notes de bas de page 5 et 6 pourraient être supprimées.

Dans la légende de la table 1, est-ce qu'il n'y aurait pas une
inversion entre Acc (décrite comme la précision) et P (décrite comme
l'exactitude).

**Resume:**

L'article s'intéresse à la révision d'articles scientifiques. Il
présente un ensemble de données issues des fichiers sources
disponibles sur arXiv. Le jeu de données contient des paires de
révision de paragraphe identifiées automatiquement à partir de LLM. Un
sous-ensemble a également été annotés manuellement par plusieurs
annotateurs. L'accord inter-annotation est modeste. Des résultats
d'un classifieur basé sur des LLM sont présentés.


Il s'agit d'un article accepté au workshop NSLP.

---

### Official Review · Reviewer_vWJe · 2026-05-06

**Mode De Presentation:** Oral

**Confience:**

Oui

**Decision:**

Accepté

**Relecture:**

L’article est bien écrit et l’approche est cohérente. À terme, on peut toutefois s’interroger sur un possible effet de "pollution" si des LLMs ont été utilisés par les auteurs pour passer de la version commentée à la version publique des paires sélectionnées.

Une brève discussion de l’accord interannotateurs serait également bienvenue. Celui-ci ne semblant pas très élevé pour une tâche qui paraît pourtant simple au premier abord. La complexité de la tâche a-t-elle été sous-estimée ? Fournir un exemple où les avis divergent serait sans doute plus intéressant que les exemples, un peu évidents, fournis en annexe.

**Resume:**

L’article présente un jeu de données portant sur les révisions de textes scientifiques. Des paires de phrases "phrase d'origine– phrase modifiée" sont extraites de fichiers LaTeX provenant d’arXiv.

La première étape de la construction consiste à extraire des blocs de texte commentés ainsi que les paragraphes similaires proches. Une annotation humaine, réalisée par cinq annotateurs sur 500 paires, permet d’obtenir un premier ensemble labellisé. Cet ensemble sert à identifier la configuration prompt–modèle la plus efficace pour réaliser la tâche de sélection des "bonnes" paires.

Le modèle sélectionné est ensuite utilisé pour construire le jeu de données final.

---

### Decision · Program_Chairs · 2026-05-07

Accept (Oral + Poster)